# Open Information Extraction via Chunks

**Kuicai Dong**[1,2], **Aixin Sun**[1], **Jung-Jae Kim**[2], **Xiaoli Li**[1,2,3]

[1] School of Computer Science and Engineering, Nanyang Technological University, Singapore
kuicai001@e.ntu.edu.sg, axsun@ntu.edu.sg
[2] Institute for Infocomm Research, A*STAR, Singapore
[3] A*STAR Centre for Frontier AI Research, Singapore
{jjkim, xlli}@i2r.a-star.edu.sg

## Abstract

Open Information Extraction (OIE) aims to extract relational tuples from open-domain sentences. Existing OIE systems split a sentence into tokens and recognize token spans as tuple relations and arguments. We instead propose *Sentence as Chunk sequence* (**SaC**) and recognize chunk spans as tuple relations and arguments. We argue that SaC has better properties for OIE than sentence as token sequence, and evaluate four choices of chunks (*i.e.,* CoNLL chunks, OIA simple phrases, noun phrases, and spans from SpanOIE). Also, we propose a simple end-to-end BERT-based model, Chunk-OIE, for sentence chunking and tuple extraction on top of SaC. Chunk-OIE achieves state-of-the-art results on multiple OIE datasets, showing that SaC benefits the OIE task. Our model will be publicly available in Github upon paper acceptance.

## 1 Introduction

Open Information Extraction (OIE) is to extract structured tuples from unstructured open-domain text (Yates et al., 2007). The extracted tuples are in the form of (*Subject*, *Relation*, *Object*) in the case of binary relations, and ($ARG_0$, *Relation*, $ARG_1$, ..., $ARG_n$) for $n$-ary relations. The structured relational tuples are beneficial to many downstream tasks, such as question answering (Khot et al., 2017) and knowledge base population (Martínez-Rodríguez et al., 2018; Gashteovski et al., 2020).

When observing benchmark OIE datasets, most relations and their arguments are *token spans*. As a domain-independent information extraction task, OIE does not specify any pre-defined extraction schema. As a result, determining the granularity or length of these text spans becomes challenging. Consequently, many existing OIE systems adopt tagging-based methods, such as BIO [1] or similar tagging schemes, to extract tuples at the *token level*.

[1] Begin, Inside, and Outside of a subject/relation/object

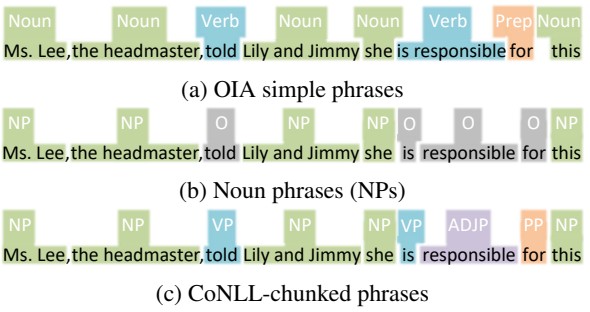

(a) OIA simple phrases

(b) Noun phrases (NPs)

(c) CoNLL-chunked phrases

Figure 1: A sentence in different chunk sequences.

Recently, Sun et al. (2020) and Wang et al. (2022) propose to use Open Information Annotation (OIA) as an intermediate layer between the input sentence and OIE tuples. OIA represents a sentence as a graph where nodes are simple phrases, and edges connect predicate nodes and their argument nodes. By employing dataset-specific rules, these OIA graphs can be transformed into OIE tuples. Nevertheless, accurately generating the complete OIA graph for a given sentence poses a challenge.

Inspired by OIA, we propose a novel notion of *Sentence as Chunk sequence* (**SaC**), as an alternative intermediate layer representation. Chunking, a form of shallow parsing, divides a sentence into syntactically related non-overlapping phrases, known as chunks (Tjong Kim Sang and Buchholz, 2000). For instance, the simple phrases in OIA can be considered as chunks (Figure 1a). To justify the adaptability of SaC for OIE, we also employ other chunking options including Noun Phrase chunks (Figure 1b) and CoNLL chunks (Figure 1c). Figure 1 shows an example sentence with different chunking schemes. Subsequently, we propose Chunk-OIE, an end-to-end tagging-based neural OIE model. Chunk-OIE performs multi-task learning among two subtasks: (i) to represent sentence in SaC, and (ii) to extract tuples based on SaC. Our findings reveal that SaC-based OIE outperforms the

traditional OIE approach representing sentences as token sequences, particularly when the OIE tuple relations and arguments align well with the chunks, as it is often the case.

Our contributions are as follows. Firstly, we propose a novel notion of Sentence as Chunk sequence (SaC) for OIE. On top of SaC, we further propose to simplify token-level dependency structure of sentence into chunk-level dependency structure in order to also encode chunk-level syntactic information for OIE. Secondly, we propose Chunk-OIE, an end-to-end learning model that (i) represents a sentence as a SaC, and (ii) extracts tuples based on the SaC. Finally, experimental results show the effectiveness of Chunk-OIE against strong baselines. Through data analysis against gold tuples, we show that chunks provide a suitable granularity of token spans for OIE.

## 2 Related Work

**OIE Systems.** OIE was first proposed by Yates et al. (2007), and TextRunner is the first system that generates relational tuples in open domain. Many statistical and rule-based systems have been proposed, including ReVerb (Fader et al., 2011), OLLIE (Mausam et al., 2012), ClausIE (Corro and Gemulla, 2013), Stanford OIE (Angeli et al., 2015), OpenIE4 (Mausam, 2016), and MINIE (Gashteovski et al., 2017). These models extract relational tuples based on syntactic structures such as part-of-speech (POS) tags and dependency trees.

Recently, two kinds of neural systems have been explored, generative and tagging-based systems (Zhou et al., 2022). Generative OIE systems (Cui et al., 2018; Kolluru et al., 2020a; Dong et al., 2021) model tuple extraction as a sequence-to-sequence generation task with copying mechanism. Tagging-based OIE systems (Stanovsky et al., 2018; Kolluru et al., 2020b; Kotnis et al., 2022) tag each token as a sequence labeling task. SpanOIE (Zhan and Zhao, 2020) uses a different approach. It enumerates all possible spans (up to a predefined length) from a sentence. After rule-based filtering, the remaining candidate spans are classified to relation, argument, or not part of a tuple. However, enumerating and filtering all possible spans for scoring is computationally expensive.

Early neural models typically seldom utilize syntactic structure of sentence, which was required by traditional models. Recently works show that encoding explicit syntactic information benefits neural OIE as well. RnnOIE (Stanovsky et al., 2018) and SenseOIE (Roy et al., 2019) encode POS / dependency as additional embedding features. MGD-GNN (Lyu et al., 2021) connects words, if they are in dependency relations, in an undirected graph and applies GAT as its graph encoder. RobustOIE (Qi et al., 2022) uses paraphrases (with various constituency form) for more syntactically robust OIE training. SMiLe-OIE (Dong et al., 2022) incorporates heterogeneous syntactic information (constituency and dependency graphs) through GCN encoders and multi-view learning. Inspired by them, we design a simple strategy to model dependency relation at the chunk level. Note that chunks in SaC partially reflect constituency structure as words in a chunk are syntactically related, by definition.

**Sentence Chunking.** Our proposed notion of SaC is based on the concept of chunking. Chunking is to group tokens in a sentence into syntactically related non-overlapping groups of words, *i.e.,* chunks. Sentence chunking is a well studied pre-processing step for sentence parsing. We can naturally use the off-the-shelf annotations as external knowledge to enhance OIE. The earliest task of chunking was to recognize non-overlapping noun phrases (Ramshaw and Marcus, 1995) as exemplified in Figure 1b. Then CoNLL-2000 shared task (Tjong Kim Sang and Buchholz, 2000) proposed to identify other types of chunks such as verb and prepositional phrases, see Figure 1c.

**OIX and OIA.** Sun et al. (2020) propose Open Information eXpression (OIX) to build OIE systems. OIX is to represent a sentence in an intermediate layer, so that reusable OIE strategies can be developed on OIX. As an implementation, they propose Open Information Annotation (OIA), which is a single-rooted directed-acyclic graph (DAG) of a sentence. Its basic information unit, *i.e.,* graph node, is a simple phrase. A simple phrase is either a fixed expression or a phrase. Sun et al. (2020) define simple phrases to be: constant (*e.g.,* nominal phrase), predicate (*e.g.,* verbal phrase), and functional (*e.g.,* wh-phrase). Edges in an OIA graph connect the predicate/function nodes to their arguments. Wang et al. (2022) extend OIA by defining more simple phrase types and release an updated version of the OIA dataset. The authors also propose OIA@OIE, including OIA generator to produce OIA graphs of sentences, and rule-based OIE adaptors to extract tuples from OIA graphs.

# 3 Methodology: Chunk-OIE

## 3.1 Task Formulation

We formulate the OIE tuple extraction process as a two-level sequence tagging task. The first level sequence tagging is to perform sentence chunking by identifying boundary and type of each chunk, and representing Sentence as Chunks (SaC). The second level sequence tagging is to extract OIE tuples on top of SaC.

Formally, given a sentence with input tokens $s_t = [t_1, \ldots, t_n]$, we first obtain the chunk sequence $s_c = [c_1, \ldots, c_m]$ $(m \leq n)$ (Section 3.2). This process can be formulated as two sequence tagging sub-tasks: (i) binary classification for chunk boundary, and (ii) multi-class classification for chunk type (See example chunk boundaries and types in the outputs of "Boundary & Type Tagging" module in Figure 2). Note that tokens at boundaries are tagged as 1 and non-boundaries as 0. Subsequently, we perform the tagging on the chunk sequence $[c_1, \ldots, c_m]$ to extract OIE tuples (Section 3.3). A variable number of tuples are extracted from a sentence. Each tuple can be represented as $[x_1, \ldots, x_L]$, where each $x_i$ is a contiguous span of chunks, either an exact match or chunk concatenation. One of $x_i$ is a tuple relation (REL) and the others are tuple arguments ($\text{ARG}_l$). For instance, the tuple in Figure 2 can be represented as (arg$_0$='Ms. Lee', rel='told', arg$_1$='Lily and Jimmy'). We address the two-level sequence tagging via multi-task learning (Section 3.4).

## 3.2 Representing Sentence as Chunks (SaC)

We first use BERT to get the contextual representations of input tokens $[t_1, \ldots, t_n]$ and then concatenate them with the POS representations to obtain the hidden representations of tokens as follows:

$$h_i = \boldsymbol{W}_{\textbf{BERT}}(t_i) + \boldsymbol{W}_{\textbf{POS}}(pos\_type(t_i)) \quad (1)$$

where $\boldsymbol{W}_{\textbf{BERT}}$ is trainable and initialized by BERT word embeddings, and $\boldsymbol{W}_{\textbf{POS}}$ is a trainable embedding matrix for POS types. The function $pos\_type(\cdot)$ returns the POS type of input token.

$h_i$ is then passed into tagging layers for chunk boundary and type classification concurrently.

$$p_i^b = \text{softmax}(h_i \cdot \boldsymbol{W}_{\textbf{bound}}^{\boldsymbol{T}} + \boldsymbol{b_1}) \quad (2)$$

$$p_i^t = \text{softmax}(h_i \cdot \boldsymbol{W}_{\textbf{type}}^{\boldsymbol{T}} + \boldsymbol{b_2}) \quad (3)$$

where $p_i^b$ and $p_i^t$ are the softmax probabilities for chunk boundary and type of token $t_i$, respectively.

Then, we chunk the sentence according to the boundary predictions, *i.e.,* the sentence is chunked to $m$ pieces if there are $m$ boundary tokens. The token is marked to be boundary if $\text{argmax}(p_i^b) = 1$. The type of each chunk is determined by the type of boundary token, which is $\text{argmax}(p_i^t)$. In overall, we convert the token sequence $[t_1, \ldots, t_n]$ into chunk sequence $[c_1, \ldots, c_m]$ by SaC.

## 3.3 SaC-based OIE Extractor

We design SaC-based OIE extractor on top of SaC. Given the typed chunks inferred by SaC (Section 3.2), we convert the BERT token representations into chunk representations, and encode the chunk types. Subsequently, we model the chunk sequence into chunk-level dependency graph. Finally, we use Graph Convolution Network (GCN) to get the chunk-level dependency graph representations. The last tagging layer performs tagging at the chunk-level to extract OIE tuples, based on the concatenation of BERT-based and GCN-based chunk representations.

**BERT-based Chunk Encoder.** The chunk representations are based on the token representations $h_i$ in Equation 1. Also, as each verb in a sentence is a potential relation indicator, verb embedding is useful to highlight this candidate relation indicator (Dong et al., 2022). We follow Dong et al. (2022) to encode tokens with additional verb embeddings, *i.e.,* $h_i^{token} = h_i + \boldsymbol{W}_{\textbf{verb}}(rel\_candidate(t_i))$, where $rel\_candidate(t_i)$ returns 1 if $t_i$ is the candidate relation indicator of the instance (otherwise, 0), and $\boldsymbol{W}_{\textbf{verb}}$ is a trainable verb embedding matrix.

For a single-token chunk ($c_i = [t_j]$), its chunk representation $h_i^{c'}$ is the same as the token representation $h_j^{token}$. For a chunk with multiple tokens ($c_i = [t_j, \ldots, t_k]$), the chunk representation $h_i^{c'}$ is the averaged token representations ($avg([h_j^{token}, \ldots, h_k^{token}])$). Moreover, we encode chunk types with a trainable chunk type embedding $\boldsymbol{W}_{chunk}$ for additional type information:

$$h_i^c = h_i^{c'} + \boldsymbol{W}_{\textbf{chunk}}(chunk\_type(c_i)) \quad (4)$$

where the function $chunk\_type(\cdot)$ returns the type (*e.g.,* Noun Phrase, Verbal Phrase) of input chunk.

**Chunk-level Dependency Graph.** Recent studies show that syntactic structures benefit neural models for NLP tasks including OIE (Fei et al.,

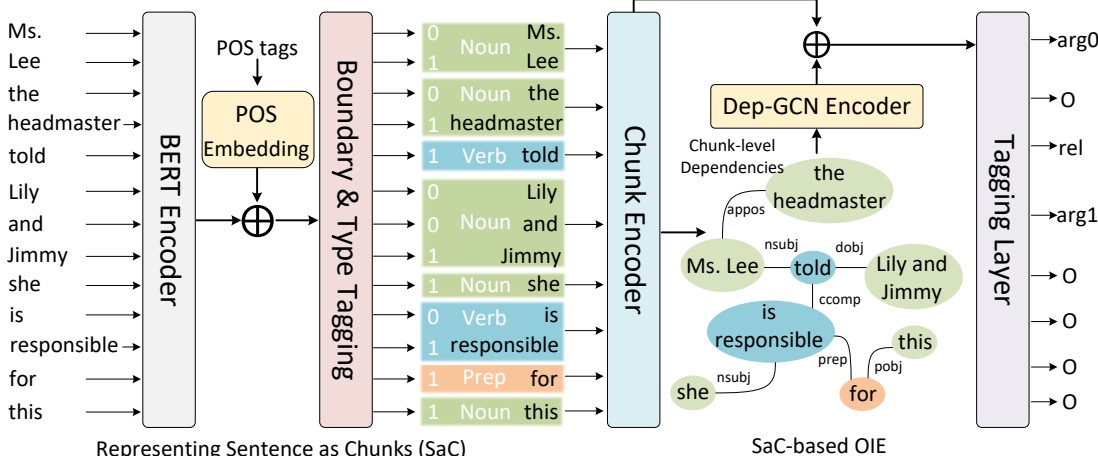

Figure 2: The overview of Chunk-OIE. Punctuation marks in the sentence are neglected for conciseness. Chunk-OIE is an end-to-end model with (i) representing Sentence as Chunks (SaC) and (ii) SaC-based OIE tuple extraction.

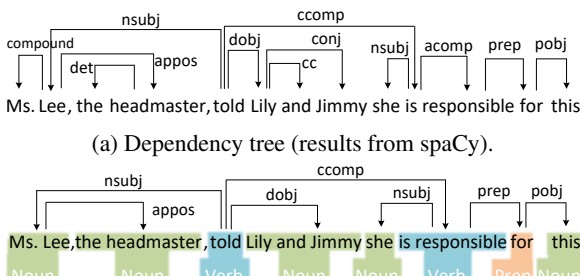

(a) Dependency tree (results from spaCy).

(b) Dependency tree at chunk level with OIA-SP.

Figure 3: Dependency trees at token-level and chunk-level (in OIA simple phrases), respectively. Note that we use spaCy to extract the dependency relations for sentences.

2021; Dong et al., 2022). Thus, given the sentence represented in SaC, we model the dependency structure of input sentence at chunk level. For this purpose, we convert a token-level dependency structure to that of a chunk level by ignoring intra-chunk dependencies and retaining inter-chunk dependencies. Figure 3 shows the chunk-level dependency tree of the example sentence in Figure 1 (with OIA simple phrases as chunks) and its dependency tree at word level.

The chunk-level dependency graph is formulated as $G = (C, E)$, where the nodes in $C$ correspond to chunks $[c_1, \ldots, c_m]$ and $e_{ij}$ in $E$ equals to 1 if there is a dependency relation between a token in node $c_i$ and a token in node $c_j$; otherwise, 0. Each node $c_i \in C$ has a node type. We label a node with the type of the dependency from the node to its parent node. Notice that *SaC greatly simplifies the modelling of sentence syntactic structure*.

**Dependency Graph Encoder.** Given the chunk-level dependency graph $G = (C, E)$, we use GCN to encode the chunk-level dependency structure. We compute the node type embedding $l_i = \boldsymbol{W_{dep}}(dep\_type(c_i))$ with a trainable matrix $\boldsymbol{W_{dep}} \in \mathbb{R}^{d_l \times N_{dep}}$, where $d_l$ is the embedding dimension and $N_{dep}$ is the number of unique dependency relations. The function "$dep\_type(\cdot)$" returns input chunk type. Subsequently, we use GCN to encode $G$ with representations as follows:

$$h_i^{dep} = \text{ReLU}\Big( \sum_{j=1}^{m} \alpha_{ij}(h_j^c + \boldsymbol{W_l} \cdot l_j + \boldsymbol{b_3}) \Big) \quad (5)$$

where $m$ refers to the total number of chunk nodes in $G$, $W_l \in \mathbb{R}^{d_h \times d_l}$ is a trainable weight matrix for dependency type embeddings, and $b \in \mathbb{R}^{d_h}$ is the bias vector. The neighbour connecting strength distribution $\alpha_{ij}$ is calculated as below:

$$\alpha_{ij} = \frac{e_{ij} \cdot \exp\big((m_i)^T \cdot m_j\big)}{\sum_{k=1}^{m} e_{ik} \cdot \exp\big((m_i)^T \cdot m_k\big)} \quad (6)$$

where $m_i = h_i^c \oplus l_i$, and $\oplus$ is concatenation operator. In this way, node type and edge information are modelled in a unified way.

For OIE extraction, we aggregate chunk representations from the BERT-based representations in Equation 4 and from the GCN-based representations in Equation 5. We then pass them into tagging layers for OIE span classification.

$$p_i^{oie} = \text{softmax}((h_i^{dep} + h_i^c) \cdot \boldsymbol{W_{OIE}^T} + \boldsymbol{b_4}) \quad (7)$$

### 3.4 Multi-task Learning Objective

As mentioned, we perform the two-level sequence tagging of sentence chunking and OIE extraction.

We combine losses from SaC and OIE tagging to jointly optimize the Chunk-OIE model.

For SaC, considering that boundary and type classification are complementary to each other, we combine the following cross-entropy losses:

$$L_{bound} = -\sum_{i=1}^{n} y_i^b \log(p_i^b) + (1 - y_i^b)\log(1 - p_i^b)$$

$$(8)$$

$$L_{type} = -\sum_{i=1}^{n}\sum_{j=1}^{c_1} y_{i,j}^t \log(p_{i,j}^t) \qquad (9)$$

$$L_{chunk} = \alpha L_{bound} + (1 - \alpha)L_{type} \qquad (10)$$

where $y^b$ and $y^t$ are gold labels for chunk boundary and type, respectively. $p^b$ and $p^t$ are the softmax probabilities for chunk boundary and type tagging obtained from Equatios 2 and 3, respectively. $c_1$ refers to the number of unique chunk types. $\alpha$ is a hyperparameter balancing the two losses.

For OIE, the gold labels are provided at token level, whereas our predicted labels are at chunk level. To enable evaluation of the generated chunk-level tuples against the token-level gold labels, we assign the predicted probability of a multi-token chunk to all its member tokens. The corresponding cross-entropy loss is computed between the predicted and the gold OIE tags:

$$L_{OIE} = -\sum_{i=1}^{n}\sum_{j=1}^{c_2} y_{i,j}^{oie}\log(p_{i,j}^{oie}) \qquad (11)$$

where $y^{oie}$ is the gold label, and $p^{oie}$ is the softmax probability obtained from Equation 7. $c_2$ is the number of unique OIE span classes.

Finally, we combine losses from Equations 11 and 12, and minimize the following multi-task learning loss:

$$L_{multi} = \beta L_{chunk} + (1 - \beta)L_{OIE} \qquad (12)$$

where $\beta$ is a hyperparameter balancing the chunking and OIE losses. More training details are in Appendix A.2.

# 4 Experiments

## 4.1 Chunk-OIE Setups

**OIE Dataset.** We conduct experiments on four datasets: the two LSOIE datasets (Solawetz and Larson, 2021), CaRB (Bhardwaj et al., 2019), and BenchIE (Gashteovski et al., 2022). More details for the OIE train/test sets are in Appendix A.6.

LSOIE is a large-scale OIE dataset converted from QA-SRL 2.0 in two domains, *i.e.,* Wikipedia and Science. It is 20 times larger than the next largest human-annotated OIE data, and thus is reliable for fair evaluation. LSOIE provides $n$-ary OIE tuples in the ($ARG_0$, $Relation$, $ARG_1$, ..., $ARG_n$) format. We use both datasets, namely LSOIE-wiki and LSOIE-sci, for comprehensive evaluation.

CaRB dataset is the largest crowdsourced OIE dataset. CaRB provides 1,282 sentences with binary tuples. The gold tuples are in the (*Subject*, *Relation*, *Object*) format.

BenchIE dataset supports a comprehensive evaluation of OIE systems for English, Chinese, and German. BenchIE provides binary OIE annotations and gold tuples are grouped according to fact synsets. In our experiment, we use the English corpus with 300 sentences and 1,350 fact synsets.

Note that the multi-task training requires both chunking and OIE labels of ground-truth. However, *chunking labels are not present in OIE datasets*. We construct chunk labels for the OIE datasets used in our experiment (in Section 4.2).

**Evaluation Metric.** For LSOIE-wiki and LSOIE-sci datasets, we follow Dong et al. (2022) to use exact tuple matching. A predicted tuple is counted as correct if its relation and all its arguments are identical to those of a gold tuple; otherwise, incorrect. For the CaRB dataset, we use the scoring function provided by authors (Bhardwaj et al., 2019), which evaluates binary tuples with token level matching, *i.e.,* partial tuple matching. The score of a predicted tuple ranges from 0 to 1. For the BenchIE dataset, we also adopt the scoring function proposed by authors (Gashteovski et al., 2022), which evaluates binary tuples with fact-based matching. A predicted tuple is counted as correct if it exactly matches to one fact tuple, and otherwise incorrect.

## 4.2 Chunk Choices and Labels Construction

SaC is to represent a sentence in *syntactically related and non-overlapping* chunks. However, there is no standard definition on what word groups should be chunked, though SaC can be achieved by any chunking scheme. We use four types of chunks to realize SaC. Also, we construct the chunking labels among OIE datasets through (i) our pre-trained chunking model or (ii) existing parser.

**CoNLL chunks.** The CoNLL-2000 (Tjong Kim Sang and Buchholz, 2000) chunking task defines 11 chunk types based on Treebank (Bies et al.,

Table 1:

| Models | LSOIE-wiki $F_1$ | LSOIE-wiki AUC | LSOIE-sci $F_1$ | LSOIE-sci AUC | CaRB $F_1$ | CaRB AUC | BenchIE $F_1$ | BenchIE $Pr$ | BenchIE $Re$ |
|---|---|---|---|---|---|---|---|---|---|
| **Token-level OIE Systems** | | | | | | | | | |
| CopyAttention (Cui et al., 2018) | 39.5[†] | 35.9[†] | 48.8[†] | 46.8[†] | 51.6[‡] | 32.8[‡] | 21.5 | 26.4 | 17.5 |
| IMoJIE (Kolluru et al., 2020a) | 49.2[†] | 47.5[†] | 58.7[†] | 55.8[†] | 53.5[‡] | 33.3[‡] | 18.4 | 38.3 | 12.1 |
| CIGL-OIE (Kolluru et al., 2020b) | 44.7[†] | 41.9[†] | 56.6[†] | 52.3[†] | 54.0[‡] | 35.7[‡] | 25.4[§] | 31.1[§] | **21.4**[§] |
| BERT (Solawetz and Larson, 2021) | 47.5[†] | 44.7[†] | 57.0[†] | 53.2[†] | 51.4[†] | 30.6[†] | 23.1 | 32.5 | 17.9 |
| BERT+Dep-GCN (Dong et al., 2022) | 48.7[†] | 47.9[†] | 58.1[†] | 55.3[†] | 52.5[†] | 32.9[†] | 25.1 | 35.3 | 19.5 |
| SMiLe-OIE (Dong et al., 2022) | 51.7[†] | 50.8[†] | 60.5[†] | 57.2[†] | 53.8[†] | 34.9[†] | 25.7 | 37.5 | 19.6 |
| **Chunk-level OIE Systems** | | | | | | | | | |
| SpanOIE (Zhan and Zhao, 2020) | 47.5 | - | 57.5 | - | 49.4[‡] | - | 23.4 | 38.1 | 16.9 |
| OIE@OIA (Wang et al., 2022) | - | - | - | - | 52.3[*] | 32.6[*] | - | - | - |
| Chunk-OIE with various SaC choice — NP$_{short}$ (2-stage) | 50.7 | 48.9 | 60.3 | 58.4 | 53.0 | 33.8 | 25.3 | 40.2 | 18.5 |
| NP$_{short}$ (end-to-end) | 51.0 | 49.1 | 60.1 | 58.8 | 52.0 | 33.5 | 25.1 | 40.5 | 18.2 |
| NP$_{long}$ (2-stage) | 48.5 | 46.4 | 57.2 | 56.7 | 50.9 | 31.7 | 23.4 | 35.1 | 17.6 |
| NP$_{long}$ (end-to-end) | 49.4 | 48.2 | 58.0 | 57.3 | 51.5 | 32.1 | 24.3 | 36.8 | 18.2 |
| OIA-SP (2-stage) | 52.1 | 50.4 | 61.2 | 60.1 | 53.6 | 35.5 | 26.7 | 41.5 | 19.7 |
| OIA-SP (end-to-end) | 52.1 | **51.0** | 61.0 | **60.4** | **54.2** | 35.2 | **27.2** | 42.4 | 20.1 |
| CoNLL (2-stage) | 52.6 | 50.2 | 60.8 | 60.2 | 53.2 | 34.7 | 26.9 | 42.0 | 19.8 |
| CoNLL (end-to-end) | **52.8** | 50.5 | **61.5** | 59.7 | 53.5 | 34.0 | 26.9 | **42.7** | 19.6 |

Table 1: Results on four OIE datasets (best scores in boldface and second best underlined). Scores with special mark are from (Kolluru et al., 2020b)[‡], (Gashteovski et al., 2022)[§], (Wang et al., 2022)[*], (Dong et al., 2022)[†].

1995). We train our own CoNLL-style chunking model, as described in Appendix A.3.

**OIA simple phrases (OIA-SP).** The OIA simple phrases has 6 types defined by Wang et al. (2022). We also train our own OIA-style chunking model, as described in Appendix A.3.

**NP chunks.** In this scheme, the tokens of a sentence are tagged with binary phrasal types: NP and O, where O refers to the tokens that are not part of any noun phrases. We notice that there often exists nested NP. Accordingly, we create two types of NP chunks, *i.e.*, NP$_{short}$ and NP$_{long}$. For example, the phrase "Texas music player" is a nested NP. NP$_{long}$ will treat it as a single NP, whereas NP$_{short}$ will split it to "Texas" and "music player" as two NPs. We use Stanford constituency parser to get NP chunks.

**SpanOIE spans.** SpanOIE (Zhan and Zhao, 2020) enumerates all possible spans of a sentence, up to 10 words. To reduce the number of candidate spans, it keeps only the spans with certain syntactic dependency patterns.

The total numbers, and average lengths of the chunks of the four types and of the gold spans of the four datasets are listed in Table 2.

### 4.3 OIE systems for Comparison

**Token-level OIE systems.** CopyAttention (Cui et al., 2018) is the first neural OIE model which casts tuple generation as a sequence generation task. IMOJIE (Kolluru et al., 2020a) extends CopyAtten-

| Spans/Chunks | Number of Spans | Average Length |
|---|---|---|
| Gold Spans | 76,176 | 4.40 |
| CoNLL | 339,099 | 1.62 |
| OIA-SP | 307,505 | 1.77 |
| NP$_{short}$ | 335,939 | 1.53 |
| NP$_{long}$ | 225,796 | 2.28 |
| SpanOIE | 1,995,281 | 4.34 |

Table 2: Number and average length of gold tuple spans, proposed phrases for SaC, and SpanOIE spans.

tion and produces a variable number of extractions per sentence. It iteratively generates the next tuple, conditioned on all previously generated tuples. CIGL-OIE (Kolluru et al., 2020b) models OIE as a 2-D grid sequence tagging task and iteratively tags the input sentence until the number of extractions reaches a pre-defined maximum. Another baseline, BERT (Solawetz and Larson, 2021), utilizes BERT and a linear projection layer to extract tuples. SMiLe-OIE (Dong et al., 2022) explicitly models dependency and constituency graphs using multi-view learning for tuple extractions. BERT+Dep-GCN is a baseline used in Dong et al. (2022), which encodes semantic and syntactic information using BERT and Dependency GCN encoder. It is the closest baseline to our Chunk-OIE. The difference is that Chunk-OIE encodes dependency structure at chunk level and the chunks partially reflect the sentence syntactic information.

**Chunk-level OIE systems.** SpanOIE (Zhan and Zhao, 2020) enumerates all possible spans from a given sentence and filters out invalid spans based on syntactic rules. Each span is subsequently scored to be relation, argument, or not part of a tuple. OIE@OIA (Wang et al., 2022) is a rule-based system that utilizes OIA graph. As the nodes of OIA graph are simple phrases (*i.e.,* chunks), we consider OIE@OIA as a chunk-level OIE system.

Chunk-OIE is our proposed model that is based on SaC for tuple extraction. To explore the effect of different chunks in SaC, we implement four variants: Chunk-OIE ($NP_{short}$), Chunk-OIE ($NP_{long}$), Chunk-OIE (OIA-SP), and Chunk-OIE (CoNLL). Besides the end-to-end Chunk-OIE proposed in Section 3, we also experiment on variants that conduct two-stage training, *i.e.,* the SaC part is pretrained with chunking dataset and frozen during the training of OIE tuple extraction (more details about 2-stage Chunk-OIE are in Appendix A.5).

### 4.4 Main Results

Experimental results in Table 1 show that Chunk-OIE, in particular its Sac-OIA-SP and SaC-CoNLL variants, achieve state-of-the-art results on three OIE datasets: LSOIE-wiki, LSOIE-sci, and BenchIE. Meanwhile, their results on CaRB are comparable with baselines. We evaluate the statistical significance of Chunk-OIE against its token-level baseline based on their $F_1$'s (each experiment is repeated three times with different random seeds). The $p$-values for Chunk-OIE (OIA-SP) and Chunk-OIE (CoNLL) are 0.0021 and 0.0027, indicating both results are significant at $p < 0.01$.

**Comparing to token-level system**: Chunk-OIE surpasses its token-level counterpart BERT+Dep-GCN on all the four datasets. Note that both Chunk-OIE and BERT+Dep-GCN rely on BERT and Dependency GCN encoder; the only difference is the input unit, *i.e.,* chunks for Chunk-OIE and tokens for BERT+Dep-GCN. Consequently, we suggest using chunks is more suitable to OIE. We observe SMiLe-OIE is a strong baseline. It explicitly models additional constituency information and the multi-view learning is computational complex. Comparing to it, Chunk-OIE is simple yet effective. CIGL-OIE performs good on CaRB dataset. It adopts coordination boundary analysis to split tuples with coordination structure, which well aligns with the annotation guidelines of CaRB dataset, but not with the guidelines of the LSOIE

| Chunk-OIE | LSOIE-wiki | | LSOIE-sci | |
|---|---|---|---|---|
| | $F_1$ | AUC | $F_1$ | AUC |
| OIA-SP | 52.1 | 50.4 | 61.2 | 60.1 |
| – w/o Dep-GCN | 51.3 | 50.2 | 59.0 | 57.8 |
| – w/o Chunk type | 50.7 | 49.8 | 59.7 | 58.1 |
| CoNLL | 52.6 | 50.2 | 60.8 | 60.2 |
| – w/o Dep-GCN | 52.0 | 49.6 | 58.4 | 58.7 |
| – w/o Chunk type | 50.4 | 49.1 | 59.8 | 58.5 |

Table 3: Ablation study of Chunk-OIE.

| Scenario | | Example |
|---|---|---|
| Match | Exact | an emissions trading system |
| | Concat. | the editor of the journal |
| Mismatch | Overlap | observed increase in trade |
| | NoOverlap | water dissolves minerals |

Table 4: Four scenarios for matching a gold tuple span (in blue) to a generated chunk (in green).

and BenchIE datasets. In Chunk-OIE, SaC treats chunks with coordination (*e.g.,* "Lily and Jimmy") as a single unit, resulting in poor scores in such cases. Except on CaRB, CIGL-OIE cannot generalize well to other datasets.

**Comparing to Chunk-level system**: Chunk-OIE with OIA-SP and CoNLL chunking schemes outperform the other two variants with $NP_{short}$ and $NP_{long}$ for all the four datasets. This indicates that multi-label chunking is more effective for the chunk-level OIE than simply recognizing noun phrases in a sentence. And, Chunk-OIE with $NP_{short}$ outperforms Chunk-OIE with $NP_{long}$ for all the four datasets, which may reflect the fact that OIE tuple arguments are often simple noun phrases rather than cascaded noun phrases. However, Chunk-OIE with OIA-SP and CoNLL chunking schemes show comparable performance.

Chunk-OIE achieves better results than SpanOIE, indicating that SaC is more reasonable than the spans enumerated by SpanOIE. Note that OIE@OIA generates tuples with rules manually crafted for OIE2006 and CaRB datasets. Also, the authors have not released source code of their rules. Therefore, OIE@OIA cannot be evaluated on LSOIE-wiki, LSOIE-sci, and BenchIE.

**Chunk-OIE: 2-stage versus end-to-end**: We notice that Chunk-OIE trained end-to-end achieves slightly better performance than Chunk-OIE with 2-stage training. This indicates that learning of sentence chunking can benefit OIE learning.

| | Chunk | CoNLL | | OIA-SP | | NP$_{short}$ | | NP$_{long}$ | | SpanOIE | |
|---|---|---|---|---|---|---|---|---|---|---|---|
| | Match Case | Percent | $L_p$ | Percent | $L_p$ | Percent | $L_p$ | Percent | $L_p$ | Percent | $L_p$ |
| **Precision** | Match | **51.0%** | 1.8 | 49.7% | 2.0 | 49.0% | 1.7 | 40.5% | 2.3 | 3.3% | 3.5 |
| | -Exact | 8.4% | 2.3 | 10.2% | 2.5 | 7.2% | 2.1 | **11.0%** | 3.4 | 3.3% | 3.5 |
| | -Concatenation | **42.6%** | 1.7 | 39.5% | 1.9 | 41.8% | 1.6 | 29.5% | 1.9 | - | - |
| | Mismatch-NoOverlap | 49.0% | 1.4 | 50.3% | 1.6 | 51.0% | 1.4 | 59.5% | 2.3 | **96.7%** | 4.4 |
| | Matching case | Percent | $L_s$ | Percent | $L_s$ | Percent | $L_s$ | Percent | $L_s$ | Percent | $L_s$ |
| **Recall** | Match | **90.5%** | 4.4 | 89.9% | 4.5 | 79.7% | 4.3 | 58.7% | 4.7 | 86.0% | 3.3 |
| | -Exact | 45.7% | 2.3 | **48.9%** | 2.5 | 37.1% | 2.1 | 36.7% | 3.4 | 86.0% | 3.3 |
| | -Concatenation | **44.8%** | 6.4 | 41.0% | 6.8 | 42.6% | 6.2 | 22.0% | 7.1 | - | - |
| | Mismatch-Overlap | 9.5% | 4.3 | 10.1% | 3.6 | 20.3% | 4.4 | **41.3%** | 4.1 | 14.0% | 12.7 |

Table 5: Precision and Recall Analysis. $L_s$ and $L_p$ are length of gold spans and generated chunks, respectively. For each type of match/mismatch case, the highest score is in boldface and second highest score is underlined.

| Candidate chunks | Precision | Recall | $F_1$ |
|---|---|---|---|
| CoNLL | **51.0** | **90.5** | **65.2** |
| OIA-SP | 49.7 | 89.9 | 64.0 |
| NP$_{short}$ | 49.0 | 79.7 | 60.7 |
| NP$_{long}$ | 40.5 | 58.7 | 47.9 |
| SpanOIE | 3.3 | 86.0 | 6.4 |

Table 6: Precision, Recall, and $F_1$ of generated chunks; best scores are in boldface, second best underlined.

## 4.5 Ablation Study

We ablate each part of Chunk-OIE (OIA-SP, CoNLL), and evaluate the ablated models on LSOIE-wiki and LSOIE-sci. The results are reported in Table 3. We first remove the dependency graph encoder. In this setting, chunking representation obtained in Equation 4 is directly used for tuple extraction. Results show that removing chunk level dependencies decreases the performance of Chunk-OIE, indicating the importance of chunk-level dependency relations. To explore the importance of chunk type, we ablate the chunk type embedding as described in Equation 4. Observe that this also leads to performance degradation.

## 4.6 Boundary Analysis on SaC

It is critical to understand the suitability of adopting chunks as the granularity for OIE. In this section, we perform boundary alignment analysis of SaC against gold spans in a benchmark OIE dataset named LSOIE. Gold Spans are the token spans of tuple arguments / relations in ground truth annotations. We analyze CoNLL chunks, OIA simple phrases, NP chunks, and SpanOIE spans as described in Section 4.2.

The boundary alignment analysis is conducted from two perspectives. (1) Precision: How often do the boundaries of SaC chunks match those of gold spans? (2) Recall: How often do the boundaries of gold spans match those of SaC chunks? There are four scenarios of boundary alignment, as exemplified in Table 4. **Match-Exact**: A gold span is exactly matched to a chunk span. **Match-Concatenation**: A gold span is mapped to multiple chunks in a consecutive sequence.[2] **Mismatch-Overlap**: A chunk overlaps with a gold span, and at least one token of the chunk is not in the gold span. **Mismatch-NoOverlap**: A chunk does not overlap with any gold span.

We show the precision and recall analysis of four boundary alignment scenarios in Table 5 and summarize the overall scores in Table 6. Observe that CoNLL chunks and OIA simple phrases show higher precision and recall of the Match boundary alignment than the other chunks. We note that the boundary alignment of CoNLL chunks to LSOIE is better than that of OIA simple phrases to LSOIE, but the two Chunk-OIE variants with CoNLL chunks and with OIA simple phrases show comparable performance. This may indicate that the precision and recall analysis of boundary alignment is 'generally good' but not 'precise' indicator for Chunk-OIE performance. We also note that SpanOIE has only 3.3% of precision, indicating that enumerating all possible spans should bear heavy burden to detect correct spans.

## 5 Conclusion

We propose Sentence as Chunk sequence (SaC) as an intermediate layer for OIE tuple extraction. We then propose Chunk-OIE, by leveraging SaC and chunk-level dependencies, achieves state-of-the-art results on several OIE datasets. We experiment on

---

[2]This is not applicable to SpanOIE, since it emulates all possible spans; if there is a match, it should be an exact match.

Chunk-OIE with various chunk choices as SaC, and perform detailed statistical study to understand to what extent these chunks align with OIE gold tuple spans, and how the boundary alignment impacts the overall OIE performance.. We indicate that CoNLL and OIA-SP chunks have better boundary alignment with OIE gold tuple spans than noun phrases, and Chunk-OIE adopting them as SaC achieves best results. In our future work, we aim to build upon SaC to develop even more effective OIE models.

## Acknowledgments

This research is supported by the Agency for Science, Technology and Research (A*STAR) under its AME Programmatic Funding Scheme (Project #A18A2b0046 and #A19E2b0098).

## Limitations

The limitations of Chunk-OIE are analyzed from three perspectives: SaC chunking errors, syntactic parsing errors, and multiple extractions issue. (**1**) Both CoNLL-chunked phrases and OIA simple phrases suffer around 10% boundary violations as shown in Table 5 (under Recall analysis). Since we use SaC as intermediate layer for OIE and perform tagging at chunk level, the chunk boundaries become a hard constraint of the extracted tuples. Among these violations, we examine 100 examples of OIA simple phrases and find that 55% of these violations are caused by chunking errors due to some complicated sentence structures. The rest is mainly caused by tuple annotation errors, meaning that all OIE systems will suffer from these annotation errors. (**2**) Chunk-OIE relies on the chunk-level dependency relations as additional syntactic knowledge. Therefore, Chunk-OIE will inevitably suffer from the noises introduced by the off-the-shelf dependency parsing tools. Also, we use POS tagger to extract all verbs in the sentence as tuple relation indicators. It is reported that the POS tagger fails to extract 8% of verbs that are suppose to be relation indicators (Dong et al., 2022). Therefore, the discrepancy between POS verbs and tuple relations may affect the OIE quality. (**3**) Moreover, there are 6% of relation indicators corresponding to multiple tuple extractions (one verb leading to more than one tuple), while our system extracts up to one tuple per relation indicator.

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

# A Appendix

## A.1 Implementation Details and Resources

We build and run our system with Pytorch 1.9.0 and AllenNLP 0.9.0 framework. The experiments are conducted with RTX 24GB 3090 GPU and Intel® Xeon® W-2245 3.90GHz CPU. Each epoch takes roughly 20 minutes for training on a single RTX 24GB 3090 GPU. We run each experiment with three random seeds and report the averaged results. We use NLP toolkit spaCy[3] to extract the POS tags and dependency relations for sentences. In addition, we obtain constituency parsing results through Stanford CoreNLP[4] and use the noun phrases to create NP-chunked phrases as part of our phrase selection exploration. The hidden dimension $d_h$ for BERT representation $h_i^{bert}$, chunked phrase representation $h_i^p$, and Dep-GCN graph representation $h_i^{dep}$ are all set to 768. The hidden dimension $d_l$ for Dep-Encoder type embedding $l_i^{dep}$ is 400.

The datasets, and their corresponding scoring scripts if applicable, used in this study are listed in Table 7. The table also list the source code URLs of the baseline models.

## A.2 Multi-Task Training

The multi-task training is elaborated in Algorithm 1. Inputs are the sentence with tokens $s_t = [t_1, \ldots, t_n]$, its chunk boundary labels $[y_1^b, \ldots, y_n^b]$,

---

[3] https://spacy.io/
[4] https://stanfordnlp.github.io/CoreNLP/

| Dataset | Resource URL |
|---|---|
| CoNLL-2000 (Tjong Kim Sang and Buchholz, 2000) | https://www.clips.uantwerpen.be/conll2000/chunking/ |
| OIA dataset v1.1 (Wang et al., 2022) | https://github.com/baidu-research/oix |
| LSOIE dataset (Solawetz and Larson, 2021) | https://github.com/Jacobsolawetz/large-scale-oie |
| CaRB dataset and scoring code (Bhardwaj et al., 2019) | https://github.com/dair-iitd/CaRB |
| BenchIE and scoring code (Gashteovski et al., 2022) | https://github.com/gkiril/benchie |

| Model | Source code URL |
|---|---|
| BERT(base-uncased) (Devlin et al., 2019) | https://huggingface.co/bert-base-uncased |
| CopyAttention (Cui et al., 2018) | https://github.com/dair-iitd/imojie |
| IMoJIE (Kolluru et al., 2020a) | https://github.com/dair-iitd/imojie |
| SpanOIE (Zhan and Zhao, 2020) | https://github.com/zhanjunlang/Span_OIE |
| CIGL-OIE (Kolluru et al., 2020b) | https://github.com/dair-iitd/openie6 |
| SMiLe-OIE (Dong et al., 2022) | https://github.com/daviddongkc/SMiLe_OIE |

Table 7: Online resources for datasets and models.

chunk type labels $[y_1^t, \ldots, y_n^t]$, and OIE span labels $[y_1^{oie}, \ldots, t_n^{oie}]$. We target to train two sub-modules $N_{SaC}$ (SaC described in Section 3.2) and $N_{OIE}$ (OIE extractor described in Section 3.3) via multi-task learning.

We first feed input tokens into $N_{SaC}$ and obtained token-level contextualized representations $h_t^s$, and the softmax probabilities $p^b, p^t$ for chunk boundaries and chunk types, respectively. We then chunk the sentence to generate SaC as $s_c = [c_1, \ldots, c_m]$. Subsequently, we pass the token-level representations $h_t^s$ and the SaC sequence $s_c$ into $N_{OIE}$. We can obtain the softmax probabilities $p^{oie}$ for OIE span predictions. For chunking, we compare the predicted chunk boundaries and types (obtained from $N_{SaC}$) with the true labels. For OIE extraction, we compare the predicted OIE spans (obtained from $N_{OIE}$) with the true spans. The cross-entropy losses for chunking and OIE extraction are combined. Finally, we calculate the gradient of the combined loss with respect to the weights of both $N_{SaC}$ and $N_{OIE}$. These gradients will be used for updating the model parameters during the training process.

### A.3 Pre-trained Chunking Model

**Model Details.** The sentence chunking is displayed in Figure 4. It is identical to the SaC part of Chunk-OIE, as described in Section 4.6. However, the training and optimization of chunking model purely relies on chunking loss in Equation 12.

**Sentence Chunking Datasets.** CoNLL-2000 Shared Task dataset by Tjong Kim Sang and Buchholz (2000) annotates 8,936 / 2,012 sentences for Train/Test sets, respectively. Open Information Annotation (OIA) v1.1 dataset by Wang et al. (2022) contains 12,543 / 2,002 / 2,077 examples for Train

---

**Algorithm 1** Multi-Task Training

**Require:**
    Input sentence $s_t = [t_1, \ldots, t_n]$,
    Chunk boundary labels $y^b = [y_1^b, \ldots, y_n^b]$,
    Chunk type labels $y^t = [y_1^t, \ldots, y_n^t]$,
    OIE span labels $y^{oie} = [y_1^{oie}, \ldots, t_n^{oie}]$.

**Ensure:**
1:  Model parameter $N_{SaC}$ and $N_{OIE}$;
2:  Initialize $N_{SaC}$ and $N_{OIE}$;
3:  **repeat**
4:      $h_t^s, p^b, p^t = N_{SaC}(s)$;
5:      $s_c = [c_1, \ldots, c_m] \leftarrow \text{chunking}(p^b, p^t)$
6:      $p^{oie} = N_{OIE}(h_t^s, s_c)$;
7:      $L_{chunk} = \text{CE}(p^b, y^b) + \text{CE}(p^t, y^t)$;
8:      $L_{OIE} = \text{CE}(p^{oie}, y^{oie})$;
9:      $L_{multi} = L_{chunk} + L_{OIE}$;
10:    Calculate the gradient;
11:    Update model parameter $N_{SaC}$ and $N_{OIE}$;
12:  **until** Stop criterion reached;
13:  **return** $N_{SaC}$ and $N_{OIE}$

---

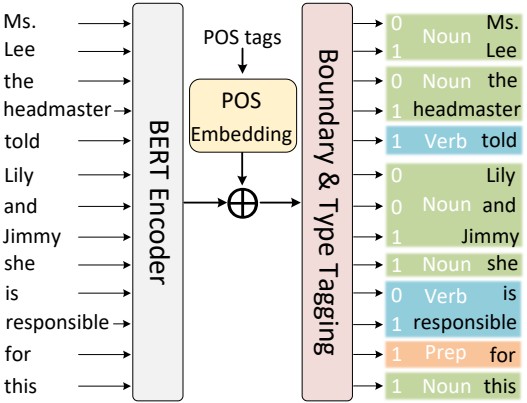

Figure 4: Sentence Chunking model. It is used to provide chunking labels for CoNLL and OIA-SP.

| Chunking model on CoNLL2000 | Chunk type $F_1$ |
|---|---|
| AT (Yasunaga et al., 2018) | 95.3 |
| Flair (Akbik et al., 2018) | 96.7 |
| MAT (Chen et al., 2020) | 97.0 |
| ACE (Wang et al., 2021) | 97.3 |
| Ours (BERT+Multi-task) | 97.0 |

Table 8: Chunk type $F_1$ on CoNLL 2000 chunking dataset. Detailed results in Appendix A.4.

| Chunking model on OIA dataset | Boundary $F_1$ | Type $F_1$ |
|---|---|---|
| Rule-based (Sun et al., 2020) | 82.4 | - |
| Neural model (Wang et al., 2022) | 88.5 | 85.3[†] |
| Ours (BERT+Multi-task) | 90.9 | 87.1 |

Table 9: Performance of chunking on OIA dataset. Note that Wang et al. (2022) report chunk boundary result only and state that 96.4% of them are labelled with correct types. We hence estimate their chunk type $F_1$ (marked with [†]) based on the given percentage.

/ Development / Test sets. Each OIA annotation is a sentence-graph pair. We only utilize the graph nodes (*i.e.,* simple phrases) for the chunking task.

**SaC Evaluation Metric.** We report Precision / Recall / $F_1$ for both chunk boundary detection and chunk type classification. For chunk boundary detection, we consider exact boundary match between a predicted chunk and a gold chunk as correct. For chunk type classification, the chunk is counted correct if both the boundary and type are exactly matched. That is, chunk type is meaningful only if its boundary is detected correctly.

**SaC (Chunking) Results.** Reported in Table 8, our SaC model is comparable to the state-of-the-art on the CoNLL-2000 dataset. Note that the best model ACE (Wang et al., 2021) leverages multiple PLMs including GloVe, fastText, ELMo, BERT, XLM-R, and XLNet, which requires significant computational resources and is slow in inference. By contrast, our model is much simpler and only utilizes BERT. On OIA dataset (see Table 9), our SaC model outperforms all the previous methods. The detailed results of chunk boundary detection and type classification, by chunk length and types, are summarized in Appendix A.4.

In overall, our proposed SaC (chunking) model achieves SOTA or comparable results in all datasets. We believe a *simple BERT-based SaC is sufficient to support our study on SaC-based OIE extraction*, as chunking is not the key focus of our study.

## A.4 Chunk Boundary and Type Analysis of Pre-trained Chunking Model

Table 10 reports the chunk boundary accuracy of our SaC model (Section 3.2) by chunk length in number of tokens. Observe that the $F_1$ of chunk boundary decreases when chunks become longer on both datasets. As expected, the longer the chunks, the harder the boundary detection becomes. Nevertheless, the $F_1$ of long chunk (*e.g.,* more than 5 tokens) is 94% and 80.44% on CoNLL-2000 and OIA datasets, respectively. This shows that our chunking model performs reasonably well in detecting long chunks. On the other hand, short chunks (*e.g.,* with 1 or 2 tokens) dominate the numbers, leading to high overall accuracy. We observe that the annotated sentences in CoNLL-2000 are longer and more formally written than that in OIA dataset. This could be a reason contributing to the higher $F_1$ on the CoNLL-2000 dataset.

Recall that chunk type classification is conditioned on the boundary provided, *i.e.,* type is meaningful only if boundary is correctly detected. If the ground truth chunk boundaries are known, the overall type classification $F_1$ is 99.2% and 95.8% respectively, on CoNLL-2000 and OIA datasets. However, in reality, the chunk boundaries have to be detected as well.

Tables 11a and 11b report the $F_1$ of chunk type classification by the major chunk types in both datasets. In this set of experiments, the chunk boundaries are detected together as type classification (*i.e.,* the same setting as in Section 3.2). In both datasets, noun, verbal, and prepositional phrases dominate the chunks. The $F_1$ scores are reasonably high on these major types. Again, as the sentences in CoNLL-2000 datasets are much longer, the number of chunks in CoNLL-2000 is much larger than that in OIA dataset, although the two datasets have comparable number of test sentences.

## A.5 Two stages Chunk-OIE model

Instead of training an end-to-end Chunk-OIE model, we also experiment on a pipeline method that consists of two-stage training, corresponding to two sub-models as shown in Figure 5. The first stage is to pre-train a SaC chunking model with chunking datasets as described in Appendix A.3. We then obtain the chunking labels for sentence in OIE datasets through the SaC sub-model. The second stage is to train the OIE extractor, during which the chunking labels are given as inputs to the

| Dataset | CoNLL-2000 | | | | OIA | | | |
|---|---|---|---|---|---|---|---|---|
| $L_{Chunk}$ | #Chunk | $Pr$ | $Re$ | $F_1$ | #Chunk | $Pr$ | $Re$ | $F_1$ |
| 1 token | 19,414 | 98.5 | 98.1 | 98.3 | 11,201 | 92.8 | 92.8 | 92.8 |
| 2 tokens | 6,267 | 97.3 | 97.4 | 97.3 | 2,924 | 88.0 | 89.5 | 88.7 |
| 3 tokens | 2,865 | 96.8 | 97.3 | 97.1 | 1,245 | 84.5 | 85.8 | 85.1 |
| 4 tokens | 945 | 94.1 | 96.4 | 95.2 | 440 | 78.3 | 78.0 | 78.1 |
| 5+ tokens | 541 | 98.7 | 90.0 | 94.1 | 421 | 95.7 | 69.4 | 80.4 |
| Overall | 30,032 | 97.8 | 97.7 | 97.8 | 16,231 | 90.4 | 91.4 | 90.9 |

Table 10: Chunk boundary extraction accuracy by chunk length.

| $Type_{chunk}$ | #Chunk | $Pr$ | $Re$ | $F_1$ |
|---|---|---|---|---|
| NP | 12,422 | 97.5 | 97.3 | 97.4 |
| VP | 4,658 | 96.7 | 96.8 | 96.7 |
| PP | 4,811 | 98.4 | 98.9 | 98.7 |
| ADVP | 866 | 88.0 | 96.0 | 87.0 |
| SBAR | 535 | 94.1 | 95.9 | 95.0 |
| ADJP | 438 | 84.8 | 93.1 | 94.0 |
| PRT | 106 | 77.9 | 89.6 | 83.3 |
| O | 6,180 | 97.7 | 97.0 | 97.4 |
| Total | 30,032 | 97.1 | 97.0 | 97.0 |

(a) CoNLL-2000

| $Type_{chunk}$ | #Chunk | $Pr$ | $Re$ | $F_1$ |
|---|---|---|---|---|
| Noun | 7,159 | 86.8 | 85.8 | 86.3 |
| Verbal | 3,673 | 83.2 | 86.3 | 84.7 |
| Prepositional | 1,517 | 91.7 | 92.5 | 92.1 |
| Logical | 811 | 75.2 | 86.9 | 80.7 |
| Modifier | 336 | 75.9 | 75.0 | 75.5 |
| Function | 60 | 37.8 | 70.0 | 49.1 |
| O | 2,675 | 96.5 | 88.3 | 92.3 |
| Total | 16,231 | 86.6 | 87.5 | 87.1 |

(b) OIA dataset

Table 11: Accuracy of chunk type classification by chunk type. Note that, for CoNLL-2000 datasets, CONJP, INTJ, LST and UCP each has fewer than 10 chunks, hence are excluded from the results.

OIE sub-model. The OIE sub-model is to train with the OIE datasets (in Section 4.1) and loss function (in Equation 11).

## A.6 Details of OIE Datasets

In this section, we elaborate more details about the train/test set of OIE datasets as mentioned in Section 4.1. For LSOIE, we follow Solawetz and Larson (2021) and Dong et al. (2022) to split the train/test set in LSOIE-wiki and LSOIE-sci domain, respectively. The statistics of LSOIE train/test sets are listed in Table 12.

CaRB only provides 1,282 annotated sentences and BenchIE provides 300 sentences, which are insufficient for training neural OpenIE models. As a result, we use the CaRB and BenchIE dataset purely for testing. We follow Kolluru et al. (2020b) to convert bootstrapped OpenIE4 tuples as labels for distant supervised model training. The statistics of CaRB and BenchIE train/test sets are listed in Table 12.

## A.7 Chunk-level Dependency Modelling

We argue that SaC simplifies the modeling of sentence syntactical structure. We elaborate this point with the example sentence shown in Figure 3a. In this sentence, "*Lee*" is the appositional modifier ('appos') of "*headmaster*". However, it is actually

| Dataset | Source | #Sent | #Tuple |
|---|---|---|---|
| LSOIE-wiki-train | QA-SRL 2.0 | 19,591 | 45,890 |
| LSOIE-wiki-test | QA-SRL 2.0 | 4,660 | 10,604 |
| LSOIE-sci-train | QA-SRL 2.0 | 38,826 | 80,271 |
| LSOIE-sci-test | QA-SRL 2.0 | 9,093 | 17,031 |
| CaRB-train | OpenIE 4 | 92,774 | 190,661 |
| CaRB-test | Crowdsourcing | 1,282 | 5,263 |
| BenchIE-train | OpenIE 4 | 92,774 | 190,661 |
| BenchIE-test | Expert | 300 | 1,350 |

Table 12: Statistics of OIE datasets used in training and evaluating Chunk-OIE.

the phrase "*Ms. Lee*" that is appositional to the phrase "*the headmaster*". If we want to model the relation between "*Ms.*" and "*the*" through token dependencies, we need to pass through three hops ('compound' → 'appos' → 'det') in order to link them up. In contrast, connecting "*Ms.*" and "*the*" via chunk-level dependencies only requires a single hop ('appos'). In another case, "*Lee*" is the nominal subject ('nsubj') and "*Lily*" is the direct object ('dobj') of verb "*told*". Apparently, we need additional dependency relations to locate the complete subject and object of "*told*". If we model dependencies at chunk level, the complete subject and object of "*told*" can be easily located to be "*Ms. Lee*" and "*Lily and Jimmy*" respectively.

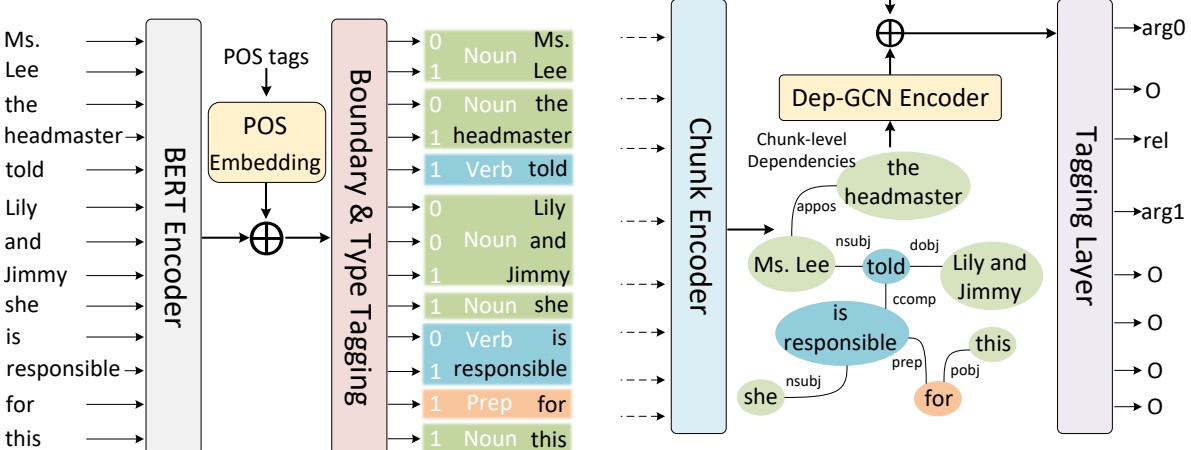

(a) Stage 1: Training SaC and extracting chunk labels

(b) Stage 2: Training OIE extractor and extracting OIE tuples based on chunks

Figure 5: The overview of Chunk-OIE with two sub-models. Punctuation marks in the sentence are neglected for conciseness. Note that stage 1 (sentence chunking) and stage 2 (OIE extraction) are trained separately.

The conversion to chunk-level dependency relations from token-level is performed in two steps. **(1)** We remove a dependency relation between two tokens if both tokens are within the same chunk. The following dependency relations in Figure 3a are removed: "compound" relation between "*Ms.*" and "*Lee*", "det" relation between "*the*" and "*headmaster*", "cc" relation between "*Lily*" and "*and*", "conj" relation between "*Lily*" and "*Jimmy*", and "acomp" relation between "*is*" and "*responsible*". **(2)** We map the remaining dependency relations, that are between tokens from different chunks, to be the relations between chunks. For example, the "appos" relation between "*Lee*" and "*headmaster*" is map to "*Ms. Lee*" and "*the headmaster*" as shown in Figure 3b. Similarly, "*Ms. Lee*" turns into the nominal subject (nsubj) and "*Lily and Jimmy*" becomes the direct object (dobj) of verb "*told*".