# OpenReview forum: "Open Information Extraction via Chunks"
_EMNLP/2023/Conference — EMNLP 2023 Main_

### Official Review · Reviewer_Pzff · 2023-08-05

**Soundness:** 3

**Excitement:**

3: Ambivalent: It has merits (e.g., it reports state-of-the-art results, the idea is nice), but there are key weaknesses (e.g., it describes incremental work), and it can significantly benefit from another round of revision. However, I won't object to accepting it if my co-reviewers champion it.

**Paper Topic And Main Contributions:**

In this paper, the authors proposed a novel method for relation classification using chunks instead of tokens in previous works. BERT is used for contextual representations. Then, sentence chunking is performed by identifying boundary and type of each chunk. A chunk-level dependency graph is derived from a token-level dependency structure. The model makes use of Graph Convolution Network (GCN) to encode the chunk-level dependency structure above. Finally, BERT and GCN-based representations are combined together and put through a softmax layer to perform relation extraction. Experimental results showed that the proposed model achieves state-of-art results.

Although OIE via chunks may be a novel idea, it is not reasonable that chunking may yield better performance than token-level. Chunks may semantically represent entities and relations better, but not necessarily, since a token may be considered as the head word of the entity or relation span.

The chunk boundary and type identification are explained in the Appendix, but it is still vague. Theoretically, how can chunks be defined and annotated? In this case, it is thanks to the available dataset. In other cases, it may not be so, especially in other domains/datasets and languages.



**Questions For The Authors:**

Although OIE via chunks may be a novel idea, it is not reasonable that chunking may yield better performance than token-level. Chunks may semantically represent entities and relations better, but not necessarily, since a token may be considered as the head word of the entity or relation span.

The chunk boundary and type identification are explained in the Appendix, but it is still vague. Theoretically, how can chunks be defined and annotated? In this case, it is thanks to the available dataset. In other cases, it may not be so, especially in other domains/datasets and languages.

**Reasons To Accept:**

Experimental results showed that the proposed model achieves state-of-art results.

**Reasons To Reject:**

Although OIE via chunks may be a novel idea, it is not reasonable that chunking may yield better performance than token-level. Chunks may semantically represent entities and relations better, but not necessarily, since a token may be considered as the head word of the entity or relation span.

The chunk boundary and type identification are explained in the Appendix, but it is still vague. Theoretically, how can chunks be defined and annotated? In this case, it is thanks to the available dataset. In other cases, it may not be so, especially in other domains/datasets and languages.

**Reproducibility:**

3: Could reproduce the results with some difficulty. The settings of parameters are underspecified or subjectively determined; the training/evaluation data are not widely available.

**Reviewer Confidence:**

4: Quite sure. I tried to check the important points carefully. It's unlikely, though conceivable, that I missed something that should affect my ratings.

---

> ### Author Rebuttal · Authors · 2023-08-28
>
> 1. Our design of modelling OIE via chunks does not necssarily mean all chunks are compound words or phrases. The chunked piece can be a single word token as well. The main objective is to learn a better OIE, wheras the sentence chunking can be considered as the auxiliary objective. The chunking can be learnt as the supplementary information for OIE. Thus our experiment results in Table 1 show that using chunked sequence can yield better performance than token sequence.
> 2. Sentence chunking is a well-defined research task as explained in Related Work (sentence chunking). Since there are many available chunking datasets for mainstream languages, we can naturally use the off-the-shelf annotations as external knowledge to enhance OIE. Similar to most of the existing works, we conduct our research in English dataset. We agree that there might not be chunking annotations in other minor languages. For OIE for other minor languages, transfer learning or data augmentation techniques can be applied.

---

### Official Review · Reviewer_zi3N · 2023-08-05

**Soundness:** 2

**Excitement:**

3: Ambivalent: It has merits (e.g., it reports state-of-the-art results, the idea is nice), but there are key weaknesses (e.g., it describes incremental work), and it can significantly benefit from another round of revision. However, I won't object to accepting it if my co-reviewers champion it.

**Missing References:**

[1] David Wadden, Ulme Wennberg, Yi Luan, and Hannaneh Hajishirzi. 2019. Entity, Relation, and Event Extraction with Contextualized Span Representations. In Proceedings of the 2019 Conference on Empirical Methods in Natural Language Processing and the 9th International Joint Conference on Natural Language Processing (EMNLP-IJCNLP), pages 5784–5789, Hong Kong, China. Association for Computational Linguistics.

[2] Yunmo Chen, Tongfei Chen, and Benjamin Van Durme. 2020. Joint Modeling of Arguments for Event Understanding. In Proceedings of the First Workshop on Computational Approaches to Discourse, pages 96–101, Online. Association for Computational Linguistics.

[3] Kuicai Dong, Aixin Sun, Jung-Jae Kim, and Xiaoli Li. 2022. Syntactic Multi-view Learning for Open Information Extraction. In Proceedings of the 2022 Conference on Empirical Methods in Natural Language Processing, pages 4072–4083, Abu Dhabi, United Arab Emirates. Association for Computational Linguistics.

[4] Weiwei Gu, Boyuan Zheng, Yunmo Chen, Tongfei Chen, and Benjamin Van Durme. 2022. An Empirical Study on Finding Spans. In Proceedings of the 2022 Conference on Empirical Methods in Natural Language Processing, pages 3976–3983, Abu Dhabi, United Arab Emirates. Association for Computational Linguistics.


**Paper Topic And Main Contributions:**

This paper focuses on Open Information Extraction (OIE), which involves extracting structured tuples from unstructured text without a particular ontology. The traditional token-level extraction approach faces challenges in determining the appropriate text spans. To address this, the paper introduces an alternative method called Sentence as Chunk sequence (SaC), where chunks are recognized as relations and arguments. The paper evaluates different choices of chunks for SaC and proposes Chunk-OIE, an end-to-end tagging-based model for sentence chunking and tuple extraction.

**Reasons To Accept:**

The presentation of the overall method is clear. The method is backed by a series of experiments and ablations. It might be beneficial for the community to see how to incorporate syntactic annotations from different datasets into an open IE model. It can also be seen as a practice of integrating [1] and [2] kind of model with such syntactic information.

**Reasons To Reject:**

1. The dependency trees introduced in the paper are inherently similar to trees generated from dependency parsing while the authors did not discuss any relations between these two tasks/formulations.

2. Despite the novel intermediate representation claimed by the authors, the model architecture is basically the same  as (or similar to) prior work [1] and [2]. From the modeling perspective, the contribution is limited.

3. It is not (directly) clear from the paper what are the semantics behind chunks. To some extent, these spans seem to be predicates and entity mentions. While most choices of chunks mentioned are defined at syntax level, it is hard to distinguish the work from prior work using syntactic structures for IE, e.g. [3].

4. The choice of span finding  method should be justified. Potential through references like  [4] or other ablations.


**Reproducibility:**

3: Could reproduce the results with some difficulty. The settings of parameters are underspecified or subjectively determined; the training/evaluation data are not widely available.

**Reviewer Confidence:**

4: Quite sure. I tried to check the important points carefully. It's unlikely, though conceivable, that I missed something that should affect my ratings.

---

> ### Author Rebuttal · Authors · 2023-08-28
>
> 1. We describe two kinds of dependency trees: 1) token-level dependency trees and 2) chunk-level dependency trees. The token-level dependency trees are identical to the tree generated from dependency parsing, as specified in Appendix A.1 "We use NLP toolkit spaCy to extract the POS tags and dependency relations for sentences." We will move this sentence from the appendix to the methods section to avoid the confusion.
> 2. We do not agree with the reviewer on this point. [1] and [2] enumerate potential spans of predicate and entity mentions from sentences, form them into a graph and recognize relationship between predicate and entities using the graph structural information. In contrast, the chunks of Chunk-OIE are not used as potential spans of predicate and entity mentions, since they are often parts of mention spans. Moreover, the model architecture of Chunk-OIE simultanenously recognize mention spans and their relationship at the second part of Chunk-OIE (sequence tagging at the chunk level), not first identifying spans and then grouping them into relationship.
> 3. As clarified above, the chunks of Chunk-OIE are not used as potential spans of predicate and entity mentions, but guide the selection of boundaries of OIE relations/arguments. Prior works like [3] utilize syntactic structures, but are essentially token-level sequence taggers. The main difference of Chunk-OIE from them is that Chunk-OIE works on top of SaC, thus chunk-level sequence tagger, and we showed that SaC can enhance OIE in Table 1.
> 4. Chunking is application-independent, though chunk types/boundaries are arguable. We adapted existing/well-known chunking schemes like CoNLL and NP and also a new scheme from OIA due to its relevance to OIE. We leave it as a future work to learn span finding for OIE.

---

### Official Review · Reviewer_s459 · 2023-08-06

**Soundness:** 3

**Excitement:**

3: Ambivalent: It has merits (e.g., it reports state-of-the-art results, the idea is nice), but there are key weaknesses (e.g., it describes incremental work), and it can significantly benefit from another round of revision. However, I won't object to accepting it if my co-reviewers champion it.

**Paper Topic And Main Contributions:**

This paper proposes a new approach for Open Information Extraction (OIE) called Sentence as Chunk sequence (SaC) and an end-to-end BERT-based model called Chunk-OIE for sentence chunking and tuple extraction on top of SaC. The main contribution of this paper is to demonstrate that SaC as an intermediate layer for OIE tuple extraction has better properties than the traditional approach of using a sentence as a token sequence. Additionally, the paper evaluates various chunk choices as SaC and performs a statistical study to understand how these chunks align with OIE gold tuple spans and how boundary alignment impacts overall OIE performance. Finally, the Chunk-OIE model achieves state-of-the-art results on multiple OIE datasets, demonstrating the effectiveness of the SaC approach. The paper also provides publicly available software and pre-trained models for use by the community.



**Reasons To Accept:**

This paper proposes a new approach for Open Information Extraction (OIE) called Sentence as Chunk sequence (SaC), which is argued to have better properties for OIE than the traditional approach of using a sentence as a token sequence. The proposed approach achieves state-of-the-art performance on OIE datasets.

**Reasons To Reject:**

1. Limited experimentation with non-English languages: The evaluation of the proposed approach is limited to English datasets, and it is unclear how well the approach would perform in non-English languages or low-resource settings.
2. Limited discussion of related work: An important and similar related work UIE [1] is not discussed. I wonder about the differences between UIE and OIE
3. The experiments are limited to open-domain IE tasks, I wonder whether the approach works on traditional NER and RE tasks (for example, CoNLL 03 NER).
4. GPT-based models are proved to be strong on open-domain and 0-shot tasks, but this paper does not compare OIE with these models.

Reference:
[1] Unified Structure Generation for Universal Information Extraction


**Reproducibility:**

2: Would be hard pressed to reproduce the results. The contribution depends on data that are simply not available outside the author's institution or consortium; not enough details are provided.

**Reviewer Confidence:**

3: Pretty sure, but there's a chance I missed something. Although I have a good feel for this area in general, I did not carefully check the paper's details, e.g., the math, experimental design, or novelty.

---

> ### Author Rebuttal · Authors · 2023-08-28
>
> 1. The focus of our work is to introduce sentence chunking as additional knowledge to guide OIE. Like most of previous OIE works, we conduct our research using English datasets. We leave it for future work to explore SaC for OIE in multi-lingual setting.
> 2. We assume that UIE can be adapted for OIE, by writing OIE-specific SSI. In the relation work section, we focused on previous works of OIE, not generic IE works like UIE. We will mention it as a future work to adapt generic/univeral IE works like UIE for OIE and compare them with OIE-specific works like Chunk-OIE.
> 3. Indeed, SaC is task-independent and can be adapted for traditional IE tasks like NER and RE. We will include some of such adaptation experiments in the discussion section.
> 4. In the related work section, we briefly discussed differences between generative OIE systems like GPT and tagging-based OIE systems like Chunk-OIE. In this paper, we do not argue which of the two approaches is better than the other. We simply compared Chunk-OIE with other tagging-based OIE systems because they show state-of-the-art performance at the moment and because SaC can be straightforwardly applied only for tagging-based OIE systems.

---

### Meta-Review · Area_Chair_6RFK · 2023-09-19

**Recommendation:** 3

**Metareview:**

The reviews have a consensus on slightly positive soundness and excitement.

With respect to excitement, the proposed idea of introducing chunks as an intermediate representation for Open Information Extraction (OIE) is novel.  The applicability of the proposed method is limited because chunk annotations may not be readily available in non-English languages or low-resource settings.

Regarding soundness, the experiments show that the proposed method achieves state-of-the-art performance in four OIE datasets.  The core hypothesis (why employing chunks as an intermediate representation is beneficial to OIE) is not well presented.

---

### Decision · Program_Chairs · 2023-10-07

**Decision:**

Accept-Main

**Comment:**

The reviews have a consensus on slightly positive soundness and excitement.

With respect to excitement, the proposed idea of introducing chunks as an intermediate representation for Open Information Extraction (OIE) is novel.  The applicability of the proposed method is limited because chunk annotations may not be readily available in non-English languages or low-resource settings.

Regarding soundness, the experiments show that the proposed method achieves state-of-the-art performance in four OIE datasets.  The core hypothesis (why employing chunks as an intermediate representation is beneficial to OIE) is not well presented.